# The Role of Thiocyanate in Modulating Myeloperoxidase Activity during Disease

**DOI:** 10.3390/ijms21176450

**Published:** 2020-09-03

**Authors:** Patrick T. San Gabriel, Yuyang Liu, Angie L. Schroder, Hans Zoellner, Belal Chami

**Affiliations:** 1Charles Perkins Centre, Faculty of Medicine and Health, The University of Sydney, Camperdown NSW 2006, Australia; psan7542@uni.sydney.edu.au (P.T.S.G.); anna.liu@sydney.edu.au (Y.L.); asch7719@uni.sydney.edu.au (A.L.S.); 2Westmead Centre for Oral Health, Sydney Dental School, The University of Sydney, Westmead NSW 2145, Australia; hans.zoellner@sydney.edu.au

**Keywords:** thiocyanate, myeloperoxidase, hypothiocyanous acid, hypochlorous acid

## Abstract

Thiocyanate (SCN^−^) is a pseudohalide anion omnipresent across mammals and is particularly concentrated in secretions within the oral cavity, digestive tract and airway. Thiocyanate can outcompete chlorine anions and other halides (F^−^, Br^−^, I^−^) as substrates for myeloperoxidase by undergoing two-electron oxidation with hydrogen peroxide. This forms their respective hypohalous acids (HOX where X^−^ = halides) and in the case of thiocyanate, hypothiocyanous acid (HOSCN), which is also a bactericidal oxidative species involved in the regulation of commensal and pathogenic microflora. Disease may dysregulate redox processes and cause imbalances in the oxidative profile, where typically favoured oxidative species, such as hypochlorous acid (HOCl), result in an overabundance of chlorinated protein residues. As such, the pharmacological capacity of thiocyanate has been recently investigated for its ability to modulate myeloperoxidase activity for HOSCN, a less potent species relative to HOCl, although outcomes vary significantly across different disease models. To date, most studies have focused on therapeutic effects in respiratory and cardiovascular animal models. However, we note other conditions such as rheumatic arthritis where SCN^−^ administration may worsen patient outcomes. Here, we discuss the pathophysiological role of SCN^−^ in diseases where MPO is implicated.

## Table of Contents

Introduction
1.1.Sources, Secretion and Elimination of SCN^−^
1.1.1.Exogenous and Endogenous Sources of SCN^−^1.1.2.Secretion and Elimination of SCN^−^
1.2.Role of MPO in SCN^−^ Biochemistry1.3.Halides and the Formation of MPO-mediated Oxidants
SCN^−^ in Diseases
2.1.Positive Effect of SCN^−^ in Disease Outcome
2.1.1.Cardiovascular Disease2.1.2.Respiratory Disease   Respiratory Viral Infections
2.2.Negative Effect of SCN^−^ in Disease Outcome
2.2.1.Smoking and Respiratory Infections2.2.2.Autoimmune Rheumatic Diseases2.2.3.Gastrointestinal Disease

Conclusions

## 1. Introduction

### 1.1. Sources, Secretion and Elimination of SCN^−^

#### 1.1.1. Exogenous and Endogenous Sources of SCN^−^

Thiocyanate (SCN^−^) is a 58 Da acidic, anionic thiolate molecule that exists in varying concentrations (0.01–2 mM) in secreted biological fluids, blood and urine [1,2,3]. There is significant variation amongst individuals with regard to SCN^−^ concentrations in physiologic fluids, and this seems due to factors such as diet and smoking habits. Non-smokers have saliva SCN^−^ concentrations of 0.5–2 mM, while concentration in smokers average around 3 mM, and some smokers may have concentrations as high as 6 mM (Table 1) [4,5,6]. One report of the SCN^−^ status of U.S. vegetarians and vegans showed that vegans have average urine SCN^−^ levels of 0.01 mM, almost double the average urine SCN^−^ levels seen in vegetarians, which was 0.006 mM [7].

Dietary sources of SCN^−^ include glucosidic cyanogen-rich plants such as almonds, cabbage, kale, broccoli, cassava, yam, maize, sugar cane, sorghum and linseed [7,8]. For example, glucobrassicin is a type of glucosinolate concentrated in cruciferous vegetables whereby plant-derived myrosinase (also known as β-thioglucosidase) mediates the hydrolysis of the glycoside, releasing a glucose and forming the unstable intermediate compound 3-indolylmethyl-isothiocyanate [9]. SCN^−^ is subsequently released at neutral pH to form the stable product, indole-3-carbinol (Figure 1). Tobacco consumption including occupationally-derived smoke intake also contributes significantly to SCN^−^ levels [10]. Further, SCN^−^ is also generated from *Pseudomonas aeruginosa* cyanogenesis, an opportunistic pathogen that infects wounds and the lungs of immunocompromised individuals [11].

Although SCN^−^ is mostly acquired from dietary sources, some is produced endogenously as a detoxification product of the reaction between cyanide (CN^−^) and thiosulfate (S_2_O_3_^2−^) in the liver [12,13] (Figure 1). The transfer of a sulfur atom between S_2_O_3_^2−^ and CN^−^ is catalysed by hepatic enzymes, including mitochondrial thiosulfate sulfurtransferase (or rhodanese) and cytosolic mercaptopyruvate sulfurtransferase (Figure 2) [14]. Sulfurtransferases are responsible for 80% of CN^−^ metabolism [12]. Additionally, oxidation of SCN^−^ into CN^−^ can be mediated by haemoglobin, with the resulting CN^−^ further detoxified by vitamin B_12_ (cobalamin) and its precursor cobinamide, before being excreted through the renal system [15,16,17].

#### 1.1.2. Secretion and Elimination of SCN^−^

Levels of SCN^−^ in the secreted fluid of mammals can vary considerably depending on numerous factors, and can reach up to millimolar concentrations for mucous membranes lining the oral cavity, digestive tract and airway [18,19].

SCN^−^ is secreted by airway, salivary, mammary, lacrimal and gastric glands, and is also present in plasma and urine. Mucosal secretions such as parotid, submandibular and whole saliva, as well as gingival crevicular fluid, dental plaque, nasal airway secretions, tears, gastric fluid and lung airway fluid can have up to approximately 2 mM SCN^−^, with saliva having the highest levels [6,18,20]. Salivary peroxidase and human lactoperoxidase (LPO) are also present in these secretions, and together with other antimicrobial defences in saliva likely account for the high levels of SCN^−^ found [4,5,21]. Airway epithelial and nasal lining secretions, on the other hand, have typically lower levels at approximately 0.5 mM SCN^−^ [18,20]. Blood plasma, breast milk and urine have SCN^−^ levels 2 to 50 orders of magnitude lower compared with the aforementioned mucosal secretions (Table 1) [21].

It was proposed that SCN^−^ is concentrated in certain fluids via energy-dependent active transport, and this was initially demonstrated in 1956 by Fletcher et al. [22]. In 1982, Tenovuo et al. showed that stimulating saliva flow rate via expectoration resulted in increased SCN^−^ concentrations when compared with unstimulated saliva collected by drooling, suggesting that SCN^−^ is actively transported in salivary glands to maintain salivary SCN^−^ levels upon increased secretion [23]. Active transport of SCN^−^ into saliva may also provide a recycling mechanism; as saliva is swallowed continuously, SCN^−^ would be reabsorbed into the blood by gastrointestinal uptake and concentrated again in salivary glands.

In human bronchial epithelium, SCN^−^ appears to be transported and concentrated via the basolaterally located sodium–iodine symporter (SLC5A5) in a Na^+^-dependent process [24]. Subsequent secretion at the apical membrane is via several separate mechanisms, including: the cAMP-mediated cystic fibrosis transmembrane conductance regulator (CFTR); purinergic agonists; Ca^2+^ and interleukin-4 (IL-4) sensitive Cl^−^ channels; the IL-4 sensitive SCN^−^/Cl^−^ exchanged pendrin (SLC26A4) [3,24,25,26].

The half-life of salivary SCN^−^ is reported to be 6–14 days [6,27,28,29,30]. While endogenous SCN^−^ does enter the glomerular filtrate, there is 90% reuptake and this accounts for the low urine SCN^−^ levels found [7,31]. Plasma SCN^−^ half-life in healthy individuals is reported to be from 1 to 5 days, and 9 days in individuals with renal insufficiency [31]. Due to its long biological half-life and the significant differences between smoker and non-smoker saliva, plasma and urinary SCN^−^ concentrations have been used as helpful biomarkers for exposure to tobacco or occupational smoke [6,10].

### 1.2. Role of MPO in SCN^−^ Biochemistry

Myeloperoxidase (MPO) is a heme homodimeric protein (~146 kDa) with functionally independent monomer units consisting of an iron protophorphyrin IX derivative located within the heavy chain of each monomer [36]. The heme unit is located within a deep cleft, restricting access of the iron atom to hydrogen peroxide (H_2_O_2_) [37]. Native MPO contains a heme unit within its active site in its ferric (Fe^3+^) form that can undergo a 2-electron oxidation reaction with H_2_O_2_, generating the highly reactive oxy-ferryl (Fe^4+^ = O) heme species containing a porphyrin π-cation radical [38], also known as Compound 1. MPO Compound 1 is highly reactive and thus can undergo 2-oxidant reduction by halides (e.g., Cl^−^, Br^−^) and pseudohalides (e.g., SCN^−^) to return to its native Fe^3+^ form [39]. This pathway is termed the “halogenation cycle”, yielding hypochlorous acid (HOCl), hypobromous acid (HOBr) and hypothiocyanous acid (HOSCN), respectively. Alternatively, Compound 1 can convert back to its native Fe^3+^ state via an independent pathway involving two sequential one-electron reductions, yielding the intermediate Compound 2 (Fe^4+^ = O) heme species in a process coined the “peroxidase cycle” [40].

The earliest report on the biological significance of SCN^−^ was in 1814 by German physician Gottfried Reinhold Treviranus, as he discovered a substance in human saliva that produced a blood-red colour when ferric ion was added [21]. In the early to mid-20th century, SCN^−^ had been of particular pharmacologic interest as an oral antihypertensive. Since then, multiple researchers have identified SCN^−^ as a potentially important factor in health and disease.

SCN^−^ plays a vital role as a substrate for human peroxidases, including MPO, LPO, salivary peroxidase, gastric peroxidase, eosinophil peroxidase and thyroid peroxidase [41]. Most of these are present in SCN^−^-containing extracellular fluids together with H_2_O_2_ [41]. These components act in concert, regulating innate immune processes as well as resident and transient flora [33].

It is important to consider the chain of events in inflammatory lesions that lead SCN^−^ to generate the potent bactericidal radical, HOSCN. Nicotinamide adenine dinucleotide phosphate (NADPH) oxidase is a multi-subunit enzyme present in neutrophils and macrophages. In inflammation, activation of NADPH oxidase catalyses the reaction between oxygen and NADPH, generating superoxide anions, a process that has been coined the “oxidative burst” [42,43]. Superoxide in turn undergoes dismutation, a process through which the anions are simultaneously oxidised and reduced to form H_2_O_2_. Dismutation can occur spontaneously or may be catalysed by the enzyme superoxide dismutase [39]. Lactoperoxidase in secretions, as well as MPO from degranulating leukocytes, catalyse the reaction of SCN^−^ with H_2_O_2_ to produce HOSCN, which is highly effective in killing microbes in these inflammatory environments through its free radical activity (Figure 3) [39].

### 1.3. Halides and the Formation of MPO-mediated Oxidants

In addition to SCN^−^, halide ions (negatively charged halogen atoms) such as Cl^−^ and Br^−^ can also be oxidised into hypohalous acids following reaction with MPO and H_2_O_2_. 

SCN^−^ has a much higher specificity for MPO than Cl^−^, with specificity constants of 1, 60, and 730 for Cl^−^, Br^−^, and SCN^−^, respectively, so that SCN^−^ is the preferred substrate for MPO [44]. However, plasma SCN^−^ levels are comparatively very low, with normal halide/pseudohalide ion concentrations in healthy human plasma at 100–140 mM (Cl^−^), 20–100 µM (Br^−^) and 200–250 µM (SCN^−^) [44].

SCN^−^ has potential to become a competitive substrate for MPO when its level is elevated beyond normal plasma level, especially in secretions where SCN^−^ levels are much higher. This is in part due to its faster reaction rate compared with Cl^−^ and Br^−^ [45]. The second-order rate constants for SCN^−^, Cl^−^ and Br^−^ are 9.6 × 10^6^ M^−1^s^−1^, 2.5 × 10^4^ M^−1^s^−1^ and 1.1 × 10^6^ M^−1^s^−1^, respectively [46].

Due to the nature and reactivity of different reactive oxygen species (ROS), HOCl is generally considered as a strong oxidant whilst HOSCN is often classified as a weak oxidant [47]. For example, the second order rate constants against cysteine residues for HOCl and HOSCN are 3.6 × 10^8^ M^−1^s^−1^ and 7.8 × 10^4^ M^−1^s^−1^, respectively. Similarly, HOSCN oxidises glutathione (GSH) with a second order rate constant considerably lower than HOCl, with values of 2.5 × 10^4^ M^−1^s^−1^ and 1.2 × 10^8^ M^−1^s^−1^, respectively [48,49].

## 2. SCN^−^ in Diseases

SCN^−^ is an important part of human host defence and health, with pharmacological interest dating back to 1857, where it was investigated for its clinical use as a hypotensive agent [50]. Use in clinical therapies is now restricted, however, due to potential toxic effects. In the 1970s, it was found that concentrations of SCN^−^ in fasting gastric juice specimens from human patients increased the likelihood of nitrosamine production and this was implicated as a potential contributor to gastric cancer [51]. Moreover, chronic toxicity of sodium thiocyanate (NaSCN) was demonstrated in the 1980s in F344 rats, while high serum concentrations of SCN^−^ were associated with lung cancer [52,53]. A more recent population-based cross-sectional study found that concentrations of urinary SCN^−^ were significantly correlated with several diseases including cancer, chronic bronchitis, emphysema, coughing, wheezing and sleep-related conditions [54]. The positive association of SCN^−^ with cancer and lung diseases in a national, population-based study supported previous investigations that utilised smaller and non-representative human sample sizes and those that used animals [54]. Recent research has focused primarily on respiratory and cardiovascular animal models of disease.

We now discuss the roles of SCN^−^ in modulating MPO in disease activity and outcome.

### 2.1. Positive Effect of SCN^−^ in Disease Outcome

#### 2.1.1. Cardiovascular Disease

Cardiovascular disease (CVD) is the primary cause of death worldwide, with the majority of CVD deaths related to coronary heart disease (CHD) [55]. CHD is characterised by chronic vascular stenosis and subsequent ischaemic injury/end-organ damage, which is primarily mediated by inflammatory remodelling of the arterial wall [56]. Endothelial dysfunction is the pre-atherosclerotic manifestation associated with invasion of immune cells into the vessel wall and the formation of ROS. It is well documented that MPO is enriched in atherosclerotic plaques and plasma concentration of MPO is a predictive factor in cardiovascular mortality, following angiography in humans [57,58].

The chlorinating activity of MPO is thought to be particularly detrimental during CVD, with high density lipoproteins and low density lipoproteins being vulnerable to oxidation by HOCl and impairing endothelial function via interference with nitric oxide production [59,60,61]. HOCl also induces endothelial apoptosis [62,63,64,65]. It was previously reported that pharmacological inhibition of MPO by 4-aminobenzoic acid hydrazide reduces plaque formation in the mouse apolipoprotein E knockout (ApoE^−/−^) model of atherosclerosis [66]. Recently, a new generation of small molecule MPO inhibitors significantly reduced the size of atherosclerotic lesion necrotic cores in Ldlr^−/−^ mice fed a western diet [67]. Despite the atherosclerotic plaque area remaining similar, MPO inhibition resulted in atherosclerotic plaque stabilisation in this murine model. Conflicting with this study is that an increase in atherosclerosis was observed in MPO^−/−^ mice [68], and this may suggest that MPO-generated reactive intermediates might be protective in murine atherosclerosis, or alternatively microbial involvement following the complete knockout of MPO, a crucial antimicrobial enzyme. The latter possibility is supported by separate work showing pro-atherogenic effects of *Porphyromonas gingivalis* in mice, rabbits and pigs [69].

High serum levels of SCN^−^ have been shown to improve long-term survival in patients following an acute myocardial infarction [70]. Ironically, smokers who are typically at risk for the early development of CVD have increased blood levels of SCN^−^ [71]. Unlike HOCl, HOSCN can be specifically degraded via thioredoxin reductase, thereby reducing its oxidative capacity in vivo [72]. In this way, HOSCN may skew the oxidative profile of MPO, thus reducing oxidative injury of the arterial wall in atherosclerosis. This is consistent with observations in ApoE^−/−^ mice fed a western diet, which have reduced atherosclerotic plaque size following 8 weeks of NaSCN treatment [73]. In this study, serum proinflammatory IL-6 levels were decreased, while IL-10 levels increased with NaSCN treatment, though no effect on monocyte or granulocyte infiltration was observed in the atherosclerotic plaque. Similarly, SCN^−^ supplementation in atherosclerosis-prone Ldlr^−/−^ mice transgenic for human MPO decreased the total plaque area with no changes to serum MPO concentrations between SCN^−^ supplemented and control mice [44]. Collectively, these studies highlight the therapeutic potential for modulating MPO oxidative activity towards the production of HOSCN in CHD.

#### 2.1.2. Respiratory Disease

SCN^−^ is extensively involved in modulating the oxidative environments of various respiratory diseases, reducing the cytotoxic effect of more powerfully oxidative HOCl. For example, Xu and colleagues demonstrated the attenuation of MPO cytotoxicity by addition of SCN^−^ to over 100 µM in the Calu-3 human lung epithelial, Neuro2a mouse neuroblastoma, human aortic endothelial cells and Min6 mouse pancreatic β cell lines [74]. In the same study, dose-dependent inhibition of MPO-produced OCl^−^ by SCN^−^ was demonstrated, with slight, partial and near complete inhibition achieved using 10 µM, 50 µM and 100–400 µM SCN^−^, respectively [74]. MPO activity in lung tissue has been previously linked to the cessation of ciliary beating [75,76] and damage to airway epithelial cells [77]. Physiologically, ciliary beating functions as a pathogen-clearing mechanism and impaired ciliary beating results in ineffective clearance of pathogenic bacterial species that can promote infection [75,78], contributing to various respiratory disorders including cystic fibrosis (CF). 

CFTR is a transmembrane receptor that functions as a chloride channel at the apical membrane of epithelial cells and mutations in CFTR results in the clinical presentation of CF [79]. Interestingly, CFTR also conducts SCN^−^, as the anion permeability of CFTR for SCN^−^ exceeds that of Cl^−^ and the concentration of SCN^−^ in the airway surface liquid is at least ~30 times its concentration in the serum [80,81]. This potentially limits harmful accumulations of Cl^−^ that subsequently form HOCl in the presence of MPO, while also facilitating the formation of the effective antibacterial compound HOSCN [82]. Unsurprisingly, many researchers have reported a deficiency in the secretion of SCN^−^ in both human CF cells [24,82], as well as human patients [20]. While nebulised hypertonic saline therapy in CF is reported to improve both SCN^−^ and GSH airway surface liquid levels, a finding which was reproduced in a CFTR knockout animal model [23,83], there are no current clinical trials evaluating nebulised SCN^−^ to date.

CF patients are particularly vulnerable to chronic *P. aeruginosa* airway infections, where the subsequent inflammatory response contributes to the major clinical problems associated with CF-lung tissue destruction [84]. Recent studies have investigated the anti-inflammatory and anti-bacterial activity of SCN^−^ through the administration of nebulised SCN^−^ in a cystic fibrosis model using beta epithelial sodium channel (βENaC) mice [85]. Compared with wild-type counterparts, βENaC mice administered SCN^−^ significantly decreased airway neutrophil infiltrate by 68%, and therefore by extrapolation, neutrophil-derived MPO, as well as rebalanced GSH redox ratio in both lung tissue and the lining fluid of airway epithelium [85]. In contrast, there was no significant effect in the reduction in elevated levels of lymphocytes and macrophages in the bronchoalveolar lavage fluid of βENaC mice by administration of nebulised SCN^−^. Moreover, SCN^−^ treatment had no effect on other cytokines, including C-X-C motif chemokine ligand 1, IL-1β, TNF-α, IFN-γ, IL-2, IL-4, IL-5, IL-6, IL-10, and IL-12 p70. Levels of the granulocyte oxidative activity biomarker glutathione sulfonamide and glutathione disulfide were also decreased in βENaC mice [85]. Interestingly, compared with wild-type mice, mean SCN^−^ levels in the epithelial lining fluid were decreased by 60% in the βENaC mice that were given vehicle treatment. Thus, the administration of SCN^−^ to βENaC mice and subsequent reduction in neutrophil infiltrate may be diverting MPO-modulated oxidative activity into another biological compartment.

Both wild-type and βENaC mice were also infected with *P. aeruginosa*, where SCN^−^ administration decreased levels of inflammation, bacterial burden, proinflammatory cytokines and 3-nitrotyrosine (only in infected wild-type mice) [85]. Bacterial burden was 70% less in wild-type mice receiving SCN^−^ as compared with wild-type mice given vehicle treatment. Furthermore, βENaC mice had an 80-fold increase in bacteria relative to their wild-type counterparts, with SCN^−^ treatment significantly reducing bacterial burden by 92% [85]. The observed increase in neutrophilic influx after *P. aeruginosa* infection suggests a consequential increase in secreted MPO, and with these data in consideration, may suggest that bacterial burden is ameliorated potentially through the microbicidal action of MPO/SCN^−^ derived HOSCN.

Taken together, the evidence suggests a protective role for SCN^−^ in respiratory airways, where SCN^−^ supplementation may provide a therapeutic effect in patients with CF. The anti-inflammatory action of SCN^−^ may potentially be attributed to a decrease in bacterial burden, as well as in the mitigation of neutrophilic infiltration, thereby diverting MPO-mediated damage away from pulmonary tissue.

##### Respiratory Viral Infections

In addition to its antibacterial activity that combats respiratory infections, SCN^−^, as well as hypothiocyanite (OSCN^−^), the conjugate base of HOSCN, holds antiviral effects that have been investigated in various in vitro influenza viral systems. OSCN^−^ virucidal activity against the A/H1N1 2009 pandemic influenza virus has been demonstrated in vitro. Specifically, a dose-dependent effect without cytotoxicity was observed where 2 µM OSCN^−^ administered achieved inhibition of viral replication by 50% prior to cell inoculation [86]. Moreover, the LPO/H_2_O_2_/SCN^−^ system has been shown to inactivate the A/Swine/02860/2009 influenza A strain virus within both differentiated rat and human tracheobronchial epithelial cells [87], and in a cell-free system [88]. Increased production of mucin and dual oxidase expression was demonstrated in the former model associated with inactivation of the influenza A virus, whilst the latter model displayed inactivation of the influenza A virus strains, including H1N1, H1N2, H3N2, and the influenza B viruses of Yamagata and Victoria lineages, though the extent of this inactivation varied between the influenza strain and LPO substrate (SCN^−^ or I^−^) [88]. Nevertheless, it appears that the antiviral capabilities of SCN^−^ and its chemical analogues working through the LPO/H_2_O_2_/SCN^−^ system are beneficially implicated in inactivating numerous influenza virus strains in vitro. In light of the current global pandemic featuring the novel severe acute respiratory syndrome coronavirus 2 (SARS-CoV-2), investigating the effects of SCN^−^ on SARS-CoV-2 in vitro and perhaps in animal models, may be a potential avenue for further research.

### 2.2. Negative Effect of SCN^−^ in Disease Outcome

#### 2.2.1. Smoking and Respiratory Infections

As tobacco smoking increases SCN^−^ levels considerably in the mucosa, one may assume that tobacco smoke can elicit antimicrobial effects; however, a plethora of evidence exists demonstrating that cigarette smoking exacerbates respiratory infections. For example, both passive and active smoking are significant risks for the development of upper respiratory tract infections, particularly otitis media [89,90,91] and for the colonisation of *Streptococcus pneumoniae* within the nasopharynx for both children and adults [92]. Furthermore, an examination of several systematic reviews and meta-analyses found that smokers are twice as likely to contract *Mycobacterium tuberculosis* infections, resulting in the development of, and death from, active tuberculosis, though interpretation may be affected due to differences in data from these studies [93,94,95]. The correlation between increased risk for the development of tuberculosis and cigarette smoke has been reported in various countries including India [96], China [97], South Africa [98] and Mexico [99]. Additionally, cigarette smoking increases the frequency of obtaining community-acquired pneumonia by approximately twofold [100] and in immunocompetent non-elderly adults, smoking is the top independent risk factor for invasive pneumococcal disease [101].

Profound changes in mucous production mechanisms and airway epithelial metaplastic changes can instead explain the increased susceptibility of smokers to respiratory infections. Squamous metaplasia of respiratory epithelium is common in habitual smokers, where specialised bronchial ciliated columnar epithelia is replaced with stratified squamous epithelia [102]. This adaptive mechanism provides a physical barrier against noxious chemicals in cigarette smoke but compromises normal ciliary function and thereby impacts the drainage of secretory products via mucociliary transport. The toxic metabolite of nicotine is also capable of significantly reducing the ciliary beat of epithelial cells as demonstrated in an in vitro model [103]. In addition, cigarette smoke increases the number and size of goblet cells in the respiratory mucosa, resulting in increased airway secretions [104]. Increased respiratory airway secretions coupled with reduced mucociliary transport is thought to increase the risk of respiratory infections, offsetting the benefit increased SCN^−^ may offer.

#### 2.2.2. Autoimmune Rheumatic Diseases

Rheumatoid arthritis (RA) is a T-cell and autoantibody-mediated autoimmune disease which results in joint damage and destruction of cartilage. Neutrophils are prominent in the pathogenesis of RA, as demonstrated in numerous RA mouse models as well as human disease [105]. In two prominent mouse models of RA, namely K/BxN antibody-mediated arthritis and collagen-induced arthritis, a specific role for MPO was indicated by reduced disease severity in MPO^−/−^ mice [106,107,108,109]. Supporting a role for MPO in human disease, the enzyme is found in both intracellular and extracellular locations in the synovium of RA patients [110]. Furthermore, levels of the MPO-mediated specific oxidation product of HOCl, 3-chlorotyrosine (3-Cl-Tyr), are significantly higher in synovial fluids of RA patients compared with those from patients with osteoarthritis [110]. Neutrophils from RA patients spontaneously generate neutrophil extracellular traps ex vivo, which are associated with MPO release and this suggests a role for leukocyte priming [111].

Destruction of cartilage, but not bone, is largely attributed to matrix metalloproteinase (MMP) activity in RA [112]. Several MMPs are implicated in the pathogenesis of RA, including MMP-1, MMP-2, MMP-3, MMP-8, MMP-9, MMP-10, MMP-12 and MMP-13, and these degrade a broad range of matrix components [112]. MMPs are produced as inactive pro-forms which require either serine-protease cleavage of an inhibitory pro-peptide domain, or oxidation of the critical thiol cysteine residue, while both scenarios expose the catalytically active Zn^2+^ site [113]. The MPO oxidation product HOCl has previously been shown to activate MMP-7 by oxidation of the key cysteine residue to a sulfinic acid form [114]. While HOCl is a relatively promiscuous oxidant, HOSCN is less reactive and highly selective for thiol sites, which represent the major site of reaction [49,115]. Therefore, it is plausible that skewing the oxidative profile of MPO from HOCl to HOSCN can result in increased oxidation of catalytic Zn^2+^ sites of pro-MMPs, increasing the overall activity of MMPs. Although the respective roles of HOSCN and MMP are not fully elucidated, it is well established that cigarette smoke and the associated increase in serum SCN^−^ is linked with increased severity and incidence of RA, and that this results in increased intensity and duration of disease, so that fewer smoking patients enter full remission [116]. In a rat model of experimental arthritis, sodium/potassium SCN^−^ was supplemented to mimic elevated levels of SCN^−^ in the blood, saliva, and urine of smokers [117]. Rats supplemented with SCN^−^ showed pro-arthritic and proinflammatory changes when subjected to various arthritic-inducing agents.

Carbamylation is a non-enzymatic post-translational modification, whereby amine or thiol groups transform into carbamyl groups via the presence of increased cyanate (OCN^−^) [118]. OCN^−^ forms a natural homeostatic equilibrium in physiological systems, where OCN^−^ concentration is too low to allow extensive carbamylation of protein. However, several environmental factors, such as smoking, cause a pathophysiological increase in OCN^−^. Increased SCN^−^ levels from cigarette smoke drive MPO Compound 1-mediated competitive oxidation of SCN^−^ to HOSCN and, to a lesser extent, OCN^−^ [71,119,120]. The conversion of lysine to homocitrulline is the most commonly described carbamylation process, and recently a new autoantibody system has been described in RA. Autoantibodies against proteins that contain homocitrulline residues (anti-carbamylated protein antibodies) are present in a subset of RA patients and can be independent from anti-citrullinated protein antibodies [119,121]. Interestingly, the presence of anti-carbamylated protein antibodies in RA patients is associated with more severe joint damage compared with patients who are negative for anti-citrullinated protein antibodies [122].

From the above, the evidence suggests that SCN^−^ is pro-arthritic via the potential pathway for MPO-derived HOSCN to activate pro-MMP at synovial sites, or the MPO/HOSCN-mediated increase in local OCN^−^ levels, increasing overall carbamylation of local proteins.

#### 2.2.3. Gastrointestinal Disease

Two main conditions drive inflammatory bowel disease (IBD) where there is chronic inflammation in the gut: ulcerative colitis (UC) and Crohn’s disease. A chronic influx of leukocytes into the gut mucosa is a major pathological presentation in IBD, which is thought to result from an abnormal host immune response to otherwise harmless commensal flora [123]. In the context of UC, the extent of neutrophil infiltration correlates with the severity of disease and is incorporated into the clinical UC severity scoring method [124]. Moreover, neutrophil depletion in rodents ameliorates experimental colitis [125,126]. Further, we earlier demonstrated a role for neutrophil-derived MPO in murine experimental colitis, with significant attenuation of disease via pharmacological inhibition of MPO [127].

It is increasingly recognised that ROS generated during inflammation plays a significant role in prolonging gastrointestinal inflammatory cycles and causing gastrointestinal injury. Colonic and faecal MPO, present in polymorphonuclear leukocytes, are significantly increased and correlate with disease severity in UC patients [128,129]. MPO is a primary source of potent ROS, including hypohalous acids HOCl, HOSCN and HOBr in the inflamed colon [130,131]. However, the proportion of MPO-oxidants formed during UC or Crohn’s disease is yet to be assessed. Despite this, 3-Cl-Tyr, a HOCl-specific biomarker, is significantly increased in colonic and serum samples of IBD patients [132].

Interestingly, active smokers who have significantly higher SCN^−^ concentrations in body fluids also exhibit reduced risk (1.7-fold) against UC and protection from the clinical symptoms including reduced flares, less need for steroids and a lower colectomy rate [133]. Thus, we speculated that SCN^−^ supplementation would confer protection during experimental colitis by redirection of MPO halogenation to favour HOSCN production above HOCl.

In our recent study, mice were supplemented with NaSCN to closely match levels in human smokers [134] before being subjected to colitis in a 3% (*w*/*v*) dextran sodium sulfate (DSS) colitis model [135]. We observed increased faecal and serum SCN^−^ levels in NaSCN-supplemented mice to above IC_50_ inhibition levels of MPO/HOCl, as determined by HOCl-mediated oxidation of luminol. Notably, 3-Cl-Tyr was found to be comparatively lower in colonic samples of SCN^−^-supplemented DSS mice. This indicated a reduction in the production of HOCl, potentially diverging MPO oxidation production to HOSCN by increased presence of free SCN^−^ ions.

However, NaSCN supplementation did not attenuate the course of experimental murine colitis. No data were collected on mouse activity levels, colon lengths, or colonic histopathology in the SCN^−^-supplemented groups. Interestingly, mice supplemented with DSS/SCN^−^ showed marked upregulation in thiol synthesis markers Nrf2 and GCLC, indicating that thiol synthesis was enhanced in this group of mice and may provide an increase in antioxidant status for the colon during DSS-insult. Overall, increasing SCN^−^ in the gut and circulation provided minimal protection against active experimental colitis.

## 3. Conclusions

SCN^−^ supplementation has been investigated in various disease models (Figure 4). Notably, in respiratory and cardiovascular disease, SCN^−^ appears to be protective against disease via direct modulation of MPO activity, favouring the production of the HOSCN oxidant. On the other hand, SCN^−^ seems to be implicated in the pathogenesis of rheumatic arthritis, while there is limited evidence to support a role in IBD. Irrespective of any possible direct roles for MPO in pathogenic mechanisms, the potential therapeutic value of SCN^−^ must be carefully considered in the context of each specific clinical condition.

## Figures and Tables

**Figure 1 ijms-21-06450-f001:**
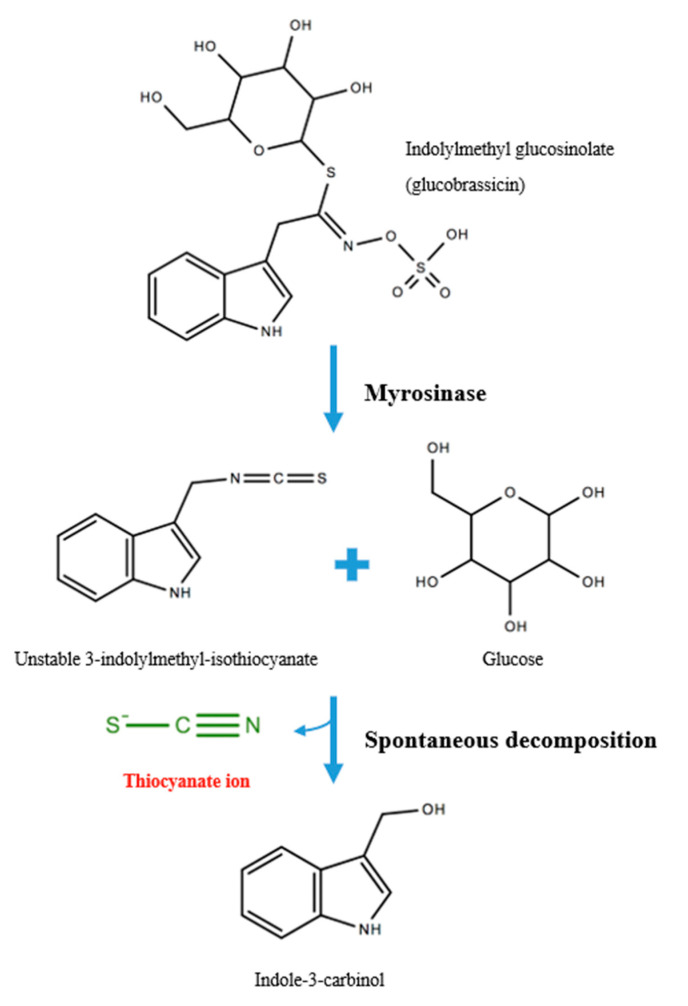
Exogenous thiocyanate (SCN^−^) production from glucosinolates.

**Figure 2 ijms-21-06450-f002:**
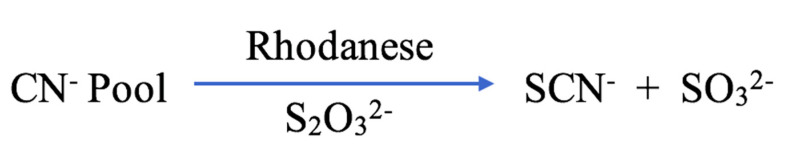
The cyanide detoxification associated endogenous SCN^−^ production pathway [12].

**Figure 3 ijms-21-06450-f003:**
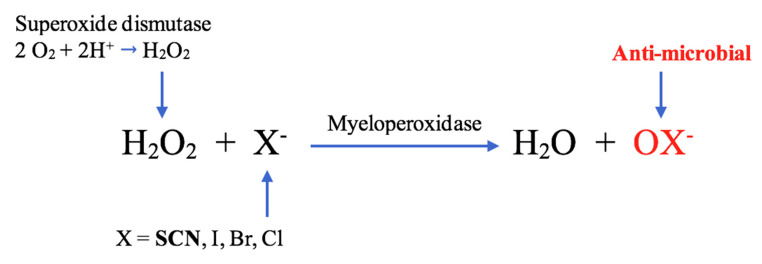
Myeloperoxidase (MPO)-mediated formation of oxidation products, including hypothiocyanous acid (HOSCN), HOI, hypobromous acid (HOBr) and hypochlorous acid (HOCl) [41].

**Figure 4 ijms-21-06450-f004:**
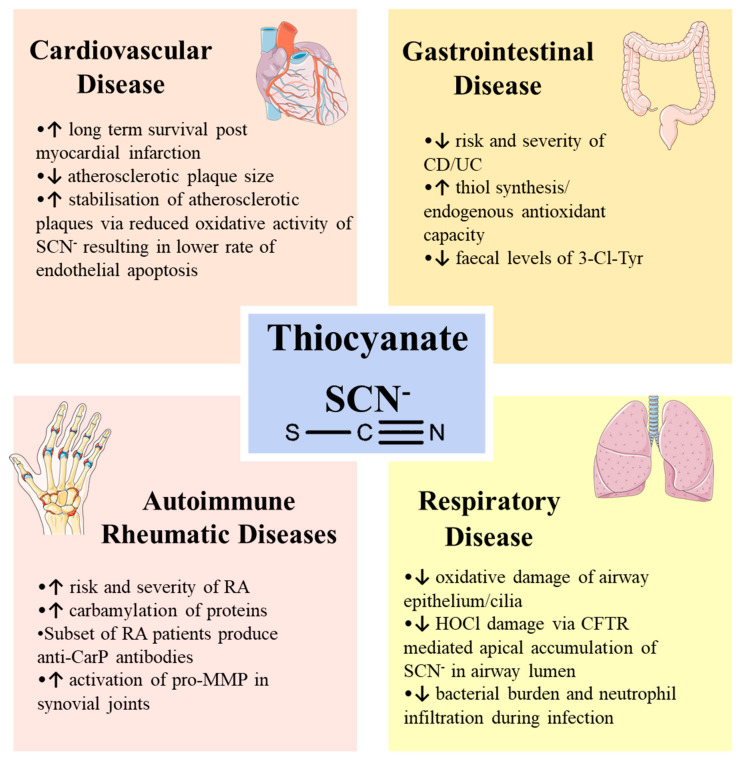
Schematic overview of SCN^−^ in various human diseases.

**Table 1 ijms-21-06450-t001:** Ranges of SCN^−^ concentration in various human biological fluids.

	Non-Smokers (mM)	Smokers (mM)	Vegan/Vegetarian (mM)	References
**Tears**	0.15	-	-	[32]
**Whole saliva**	0.5–2	2–3.6	-	[4,5,6]
**Nasal airway fluids**	1–1.2	-	-	[33]
**Lung airway fluid**	0.03–0.65	-	-	[18,20]
**Breastmilk**	0.0001–0.004	-	-	[34,35]
**Gastric fluids**	0.25–0.3	-	-	[2]
**Plasma**	0.03–0.05	0.1–0.2	-	[4,6]
**Urine**	0.009–0.024	0.33–0.275	0.002–0.05/0.001–0.034	[7,10]

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
