# Peer review of "The Role of Thiocyanate in Modulating Myeloperoxidase Activity during Disease"

_ijms, 2020, doi:10.3390/ijms21176450_

Round 1
Reviewer 1 Report
Please see the attached file.

Author Response
We would like to thank the reviewers for providing insightful feedback. Please see below our responses and references to the manuscript where we made changes. References to amendments and their page and line number(s) are highlighted in pink in this document.
Reviewer 1 (Changes on manuscript highlighted in blue)
This is an interesting scoping review on the potential we discuss the therapeutic potential use of SCN to influence disease outcomes in various pathological conditions.
Please address the following revisions:
1) The evidence discussed by the authors could be more structured by in vitro, in vivo and animal studies.
We have structured our review by diseases and as per recommendation by Reviewer 3, we have further sub-categorised diseases by the potential benefit or disadvantage thiocyanate may play. We feel to create another subcategory for in vitro and in vivo studies will make the structure of the manuscript unnecessarily complicated.
2) When you say “Cardiovascular disease (CVD) is the primary cause of death worldwide, with the majority of CVD deaths related to coronary heart disease (CHD)”, that is not true for several Western countries. Due to cardio-vascular health promotion programs cancer is the leading cause of deaths in several High income countries.
While it is true that some countries in the Western world (e.g. Canada) have cancer as leading cause of death, the majority of Western countries (USA, UK, Australia et cetera) report cardiovascular disease (CVD) as a leading cause of death and the latest statistics (2020) from the World Health Organisation (WHO) also reports ischaemic heart disease (i.e. CVD) is the leading cause of death worldwide. As such, we feel the statement is correct although we acknowledge some countries may have other leading causes of death.
3) When discussing Respiratory Disease, the anti-viral, anti-bacterial and anti-fungal activity of OSCN- could be discussed, citing also relevant studies in vitro. This is part of the beneficial/protective effect of OSNC- against respiratory infections.
Cegolon, L., Salata, C., Piccoli, E., Juarezc, V., Palu’, G., Mastrangelo, G., Calistri, A., 2014. In vitro antiviral activity of hypothiocyanite against A/H1N1/2009 pandemic influenza virus. Int. J. Hyg Environ. Health 217, 17–22.
Patel, U., Gingerich, A., Widman, L., Demba, S., Tripp, R.A., Rada, B., 2018. Susceptibility of influenza viruses to hypothiocyanite and hypoiodite produced by lactoperoxidase in a cell-free system. PLoS One 13 (7). https://doi.org/10.1371/journal.pone. 0199167. In this issue.
Cegolon L. Investigating hypothiocyanite against SARS-CoV-2. International Journal of Hygiene and Environmental Health 227 (2020) 113520
Gingerich A, Pang L, Hanson J, Dlugolenski D, Streich R, Lafontaine ER, Nagy T, Tripp RA, Rada B. Hypothiocyanite produced by human and rat respiratory epithelial cells inactivates extracellular H1N2 influenza A virus. Inflammation Research; 65: 71 – 80.
Thomas EL, Aune TM. Lactoperoxidase, peroxide, thiocyanate antimicrobial system: correlation of sulfhydryl oxidation with antimicrobial action. Infect. Immun. 1978; 20: 456 – 463.
Conner GE, Wijkstrom-Frei C, Randell SH, Fernandez VE, Salathe M. The lactoperoxidase system links anion transport to host defense in cystic fibrosis. FEBS Lett. 2007; 581: 271 – 278.
We have now created a subsection (2.1.2.1. Respiratory Viral Infections) (page 7, line 262) under Section 2.1.2. Respiratory Disease, which discusses the anti-viral activity of OSCN-.
4) More discussion around beneficial/detrimental effect of cigarette smoking on risk of respiratory infections could be provided.
We have now included a discussion on the effect of cigarette smoking on risk of respiratory infections (page 8, lines 281-306) within the new subsection 2.1.2.1. Respiratory Viral Infections (page 7, line 262).
Reviewer 2 Report
This is a review of the literature on the subject of thiocyanate and its role in disease. Reviewing this subject is not a straightforward task to undertake as thiocyanate exists in different body fluids and locations with wide ranging concentration and likely its associated physiological roles also. The authors have made a promising start in summarising the studies investigating thiocyanate in different disease groups. I feel that at this stage the manuscript would benefit from some editing to increase the accuracy of interpretation of previous studies and to clarify the overall message of the review. I have noted a few points that I hope will help with developing this into a useful review.
- Are the mM concentrations of thiocyanate in Table 1 means or ranges? The values discussed in the text below the table are somewhat different. Also it might be useful to convert the µg/L to mM.
- In section 1.3 it would be best to avoid making a broad statement about ROS in general as they vary widely in their potency as oxidants and as anti-microbial agents.
- I would suggest reconsidering the title of the review. There is very little about the chemistry of myeloperoxidase which really warrants a section for its introduction. Figure 3 is oversimplified.
- There is a tendency to over-reference reviews, primary references should be used for data.
- An effort should be made to include the most recent studies published on thiocyanate and hypothiocyanous acid.
- I would recommend a recent review that would help direct the authors: The Role of Myeloperoxidase in Biomolecule Modification, Chronic Inflammation, and Disease. In ARS Vol 32, 2020, by Davies and Hawkins.
- Also some of the references used seem inappropriate, and I think may have led to some misunderstanding. For example the oxidative burst actually refers to neutrophils’ massive consumption of oxygen by the NADPH oxidase - the way it is written in section 1.4 is not exactly wrong but it is unclear.
- For completeness, the formulae of cyanide and cyanate should be included.
- There are some inconsistencies, for example with statements about HOSCN as an oxidant and bactericide. It would be good to give the thermodynamic properties/constants, with units. In the abstract HOSCN is a “less potent species”, in section 1.5 it is a “weak oxidant”, and in section 2.4 it is a “potent antibacterial compound”.
- The section numbering jumps from 2.4 to 5.
Author Response
Reviewer 2 (Changes on manuscript highlighted in green)
Are the mM concentrations of thiocyanate in Table 1 means or ranges? The values discussed in the text below the table are somewhat different. Also it might be useful to convert the µg/L to mM.
Table 1 reflects the ranges which we have now explicitly stated (page 4, line 100). We have converted all instances of ug/mL to mM. (page 1, lines 36 & 37). With regards to the reviewer’s comment of somewhat varying ranges of SCN mentioned in text to the table, we have reviewed the referenced literature and are confident with the ranges listed however we have narrowed the range of SCN in non-smokers’ saliva from 0.2-2 mM to 0.5 -2 mM (page 4, line 100 – Table 1) in accordance to the reference Minarowski, L., et al., 2008, Folia Histochemica et Cytobiologica.
In section 1.3 it would be best to avoid making a broad statement about ROS in general as they vary widely in their potency as oxidants and as anti-microbial agents.
As per reviewer 3’s comments, we have removed this section (page 4, lines 91-100, in previous manuscript version) as it was overly generalised and did not contribute to the direction of the review.
I would suggest reconsidering the title of the review. There is very little about the chemistry of myeloperoxidase which really warrants a section for its introduction. Figure 3 is oversimplified.
We appreciate the view of the reviewer and agree that there is very little in the way of MPO chemistry in this review. This manuscript is disease-centric and as such we feel the title does not reflect a chemical analysis but rather a disease-focus approach.
We have included more detail of MPO structure and catalytic activity in Section 1.3 (page 4, lines 103-115).
Figure 3 is purposefully not MPO-centric and although simplistic, it adequately demonstrates the production of HOSCN from the SCN psuedohalide, catalysed by MPO. The 'take home' message is that H2O2 generated from 'oxidative burst' is consumed in order to generate (pseudo)-hypohalous acids.
There is a tendency to over-reference reviews, primary references should be used for data.
We have reassessed 135 of our references and found 6 were review articles. We have replaced 2 of the 6 review articles with primary references where they supported data directly.
- Reference #19 (Chandler JD, Day BJ, 2015, Free Radical Research) was used in Section 2.1.2. Respiratory Disease (page 7, lines 234, 237, 242, 249, 252) with a new reference – now Reference #85 (Chandler JD et al., 2015, Am J Respir Cell Mol Biol).
- Reference #45 (Pattison et al., 2012, Free Radical Research) was used (page 5, line 148) in Section 1.3 Halides and the Formation of MPO-mediated Oxidants with a new reference – now Reference #46 (Furtmuller et al., 1998, Biochemistry).
The remaining 4 review articles were used to support conceptual claims and we felt this was appropriate.
An effort should be made to include the most recent studies published on thiocyanate and hypothiocyanous acid. I would recommend a recent review that would help direct the authors: The Role of Myeloperoxidase in Biomolecule Modification, Chronic Inflammation, and Disease. In ARS Vol 32, 2020, by Davies and Hawkins.
We have evaluated our references and studies available in the niche research area of thiocyanate modulation of myeloperoxidase and are confident that we have utilised most recent studies published in the disease categories we described, save for the reference below which we have now added https://doi.org/10.1016/j.abb.2020.108490 (page 10, line 380) as Reference #135.
We have now included a new subsection (page 7, lines 262-306) for the role of thiocyanate in viral respiratory infections and have utilised recent studies in relation to this new subsection, as per Reviewer 1’s suggestions.
As the focus of this review is SCN & MPO in various diseases, we have not included recent or otherwise original manuscripts of thiocyanate & MPO modulation where they are not modelled in a disease setting (i.e., basic science/chemistry).
Also some of the references used seem inappropriate, and I think may have led to some misunderstanding. For example the oxidative burst actually refers to neutrophils’ massive consumption of oxygen by the NADPH oxidase - the way it is written in section 1.4 is not exactly wrong but it is unclear.
We have addressed this by placing a more appropriate primary reference, as well as a review (page 4, line 129) and have modified the text as below (page 4, lines 127-129) to reflect the process of superoxide generation via oxidative burst.
'Nicotinamide adenine dinucleotide phosphate (NADPH) oxidase is a multi-subunit enzyme present in neutrophils and macrophages. In inflammation, activation of NADPH oxidase catalyses the reaction between oxygen and NADPH, generating superoxide anions, a process that has been coined the ‘oxidative burst’ [42, 43]. Superoxide in turn undergoes dismutation, a process through which the anions are simultaneously oxidised and reduced to form H2O2. Dismutation can occur spontaneously or may be catalysed by the enzyme superoxide dismutase [39] '.
For completeness, the formulae of cyanide and cyanate should be included.
Cyanide was previously already abbreviated in-text (page 1, line 31 in previous manuscript version) although we have since removed the Section that contained this and have moved the first instance of using ‘cyanide’ and its abbreviation to Section 1.1.1. Exogenous and Endogenous Sources of SCN‑ (page 2, line 48) with the other instances where CN- was also used (page 2, lines 49, 52, 53 in current manuscript version) and in the abbreviations list (page 12, line 411).We have now modified one instance where ‘cyanide’ was used instead of its abbreviation (page 2, line 51) We have now also replaced ‘cyanate’ with its formula ‘OCN-‘ in-text (page 9, lines 337, 338, 340, 342, 351) and in the abbreviations list (page 12, line 425).
There are some inconsistencies, for example with statements about HOSCN as an oxidant and bactericide. It would be good to give the thermodynamic properties/constants, with units. In the abstract HOSCN is a “less potent species”, in section 1.5 it is a “weak oxidant”, and in section 2.4 it is a “potent antibacterial compound”.
Thank you for bringing this important point to our attention. It is important to note that the potency of HOSCN as an oxidant is referred to against HOCl which we failed to emphasise in the text. We have made several changes in the text and abstract that now reflect this. Specifically, the oxidative capacity of HOSCN to oxidise cysteine residues is significantly reduced when compared to HOCl and in this capacity we refer to HOSCN as a weaker oxidant. However, it is incorrect to assume that HOSCN is a weak antibacterial compound based on the above and we have cited original research articles that demonstrate the effectiveness of HOSCN in killing bacterial species. In this regard, we have changed 'potent' to 'effective' (page 7, line 224) and added ‘a less potent species relative to HOCl’ in the abstract (page 1, lines 19-20).
We have also elaborated on the thermodynamic properties/constants as follows:
'For example, the second order rate constants against cysteine residues for HOCl and HOSCN are 3.6 x 108 k2/M-1s-1 and 7.8 x 104 k2/M-1s-1, respectively. Similarly, HOSCN oxidises glutathione with a second order rate constant considerably lower than HOCl, with values of 2.5 x 104 k2/M-1s-1 and 1.2 x 108 k2/M-1s-1, respectively [48, 49].' (page 5, lines 150-154) to support the previous sentence in the paragraph where ‘HOCl is generally considered as a strong oxidant whilst HOSCN is often classified as a weak oxidant [47]’.
The section numbering jumps from 2.4 to 5.
We have now corrected the section numbering and '5. Conclusions' now reads '3. Conclusions' (page 12, line 394).
Reviewer 3 Report
Manuscript details:
Journal: International Journal of Molecular Sciences (IJMS)
Manuscript ID: ijms-869623
Type of manuscript: Review
Title: The role of thiocyanate in modulating myeloperoxidase activity during disease
Comments and Suggestions for Authors
Article review
In this submission on the matter of SCN- anion which modulation MPO activity, the authors present the current knowledge in this topic based on a review of publications (110 articles in references). The authors provide an interesting summary of the topic of the thiocyanate ion and its relationship to MPO as well as the potential therapeutic properties of this anion.
The article is well organized (chapter one is about SCN- and halides and their relationships to MPO, chapter two deals with inter alia administration of SCN- in modulating MPO activity (animal models) and in association with various diseases. Below, I present my main comments.
Notes to chapter one:
1.1. SCN- Molecular Structure - this section mixes the molecular structure of this ion (very little information) with the presence of the ion in various biological fluids and at different concentrations. It should only target the molecular structure.
1.2. Sources and Elimination of SCN- - the division of this subsection into two, i.e. sources of occurrence (exogenous and endogenous) and the subsection on ion elimination - such a division will be clearer.
1.3. The Anti-Microbial Activity of Free Radicals - this subsection does not provide relevant information related to the topic of the article, its removal is suggested.
1.4. Redox Biochemistry of SCN- and HOSCN - does redox biochemistry apply to SCN- and HOSCN? if it is rather Redox Biochemistry of SCN- and role HOSCN. But taking into account the title of the article and the content of this subsection, the name "Role of MPO in SCN- biochemistry" would be more accurate.
1.5. Competitive ROS Formation – but in the range of halides, which should be emphasized in the name of the subsection
Mirosinase (β-thioglucosidase, EC 3.2.3.1) is an enzyme found in Brassicaceae that catalyzes the breakdown of glucosinolates. The enzyme is present in plant cells, hence its action is possible after crushing the tissues and releasing the juice from. The enzymatic breakdown of thioglycosides therefore takes place in the mouth when chewing food, and also when grinding vegetables during food preparation. The temperature of 90oC causes denaturation and inhibits the enzyme activity. Thiocyanates are one of the products of enzymatic hydrolysis of glucosinolates (nitriles, isothiocyanates, indol are also produced). Hence, Figure 2 is unclear to the reader, is it really endogenous SCN-?
In the text is S2O3- it should be S2O3-2 (1.2. Sources and Elimination of SCN- - third row).
Notes to chapter two
2. SCN- in Disease - this chapter deals with various diseases so it is more accurate to call it SCN- in Diseases. Thus, it will be clearer to divide it into two sections - diseases in which SCN- ions have a positive effect, and the second subsection - diseases in which SCN- ions have a harmful effect. Figure 4 will summarize this. In this chapter, it is also worth paying attention to the effect of SCN- ions on the thyroid (although in the context of MPO it may be difficult to relate). Thiocyanates present in food are natural anti-nutritional substances, they easily penetrate the cell membranes, disrupt the absorption of iodine by the thyroid gland, and disturb the synthesis of thyroid hormones, they are natural goiter-forming substances. Frequent consumption of cruciferous vegetables, which contain thioglycosides transformed, inter alia, into thiocyanate, is an important element in the prevention of cancer but requires the simultaneous supply of products with a high iodine content.
Quotation - Levels of SCN- in the extracellular fluid of mammals can vary considerably depending on numerous factors, and can reach up to millimolar concentrations for environmentally exposed mucous membranes lining the oral cavity, digestive tract and airway [16, 49]. (last paragraph chapter 2) - what extracellular fluid do we mean? what is this environmental exposure all about?
GSH is missing from the abbreviations
I suggest considering carefully the first three sentences of the Abstract; my suggestion (below) and abstract in the article.
Thiocyanate (SCN-) is a pseudohalide anion omnipresent across mammals and is particularly concentrated in secretions within the oral cavity, digestive tract, and airway. Thiocyanate can outcompete chlorine anion (was - free chlorine - whether free chlorine = chloride ions?), and with other halides (Br-, J-, F-) as substrates for myeloperoxidase by undergoing two-electron oxidation with hydrogen peroxide. This forms its respective hypohalous acids (HOX where X- = halides) and in the case of thiocyanate HOSCN, which is also a bactericidal oxidative species involved in the regulation of commensal and pathogenic microflora.
The last sentence of the Abstract - Here, we discuss the therapeutic potential of SCN- for altering disease outcomes across various pathologic processes. My question - why have diseases in which SCN- has no therapeutic effect been also described?
Author Response
Reviewer 3 (Changes on manuscript highlighted in yellow)
1.1. SCN- Molecular Structure - this section mixes the molecular structure of this ion (very little information) with the presence of the ion in various biological fluids and at different concentrations. It should only target the molecular structure.
As this section has very little information, we have removed this section entirely (page 1, lines 27-34 in previous manuscript version) and moved the contents elsewhere, where relevant (page 1, line 31-32).
1.2. Sources and Elimination of SCN- - the division of this subsection into two, i.e. sources of occurrence (exogenous and endogenous) and the subsection on ion elimination - such a division will be clearer.
We have created new subheadings and renamed this section (previously Section 1.2; now Section 1.1) as follows:
1.1 - Sources, Secretion and Elimination of SCN- (page 1, line 28)
1.1.1 - Exogenous and Endogenous Sources of SCN- (page 1, line 29)
1.1.2 - Secretion and Elimination of SCN- (page 3, line 61)
All subsequent sections have been adjusted accordingly to align with changes in text structure within the manuscript in accordance with all reviewers’ comments.
1.3. The Anti-Microbial Activity of Free Radicals - this subsection does not provide relevant information related to the topic of the article, its removal is suggested.
We have removed this section according to reviewer 2's comment. (page 4, lines 91-100, in previous manuscript version)
1.4. Redox Biochemistry of SCN- and HOSCN - does redox biochemistry apply to SCN- and HOSCN? if it is rather Redox Biochemistry of SCN- and role HOSCN. But taking into account the title of the article and the content of this subsection, the name "Role of MPO in SCN-biochemistry" would be more accurate.
We have considered the above comment and agree that the redox biochemistry of HOSCN is not discussed. Thus, we have amended the subheading to 'Role of MPO in SCN- biochemistry', which is now section 1.2 after the removal of the aforementioned section.
(page 4, line 102)
1.5. Competitive ROS Formation – but in the range of halides, which should be emphasized in the name of the subsection
We have made the subheading of this section (now section 1.3) more specific to the contents and nature of the manuscript – now entitled: 'Halides and the Formation of MPO-mediated Oxidants' (page 5, line 138).
Mirosinase (β-thioglucosidase, EC 3.2.3.1) is an enzyme found in Brassicaceae that catalyzes the breakdown of glucosinolates. The enzyme is present in plant cells, hence its action is possible after crushing the tissues and releasing the juice from. The enzymatic breakdown of thioglycosides therefore takes place in the mouth when chewing food, and also when grinding vegetables during food preparation. The temperature of 90oC causes denaturation and inhibits the enzyme activity. Thiocyanates are one of the products of enzymatic hydrolysis of glucosinolates (nitriles, isothiocyanates, indol are also produced). Hence, Figure 2 is unclear to the reader, is it really endogenous SCN-?
Figure 2 was unfortunately mislabelled and should read 'Exogenous SCN- production from glucosinolates'. We have now made this correction for the figure title (page 2, line 56). After amending subsections of Section 1.1, we have reshuffled Figure 2 earlier in the text and is now Figure 1. Figure 1 links to the paragraph above in the manuscript, which very briefly describes dietary sources of glucosidic compounds. We have now expanded this paragraph to describe more detail on the myrosinase-mediated conversion of glucosinolates into SCN-. (pages 1-2, lines 39-43). We have made changes to Figure 1 itself to simplify the conversion of glucosinolates to thiocyanate (page 2, line 55).
In the text is S2O3- it should be S2O3-2 (1.2. Sources and Elimination of SCN- - third row).
We have made the correction for this in the text (page 2, lines 48 & 49) and in the abbreviations list (page 12, line 402).
- SCN-in Disease - this chapter deals with various diseases so it is more accurate to call it SCN-in Diseases. Thus, it will be clearer to divide it into two sections - diseases in which SCN- ions have a positive effect, and the second subsection - diseases in which SCN- ions have a harmful effect. Figure 4 will summarize this. In this chapter, it is also worth paying attention to the effect of SCN- ions on the thyroid (although in the context of MPO it may be difficult to relate). Thiocyanates present in food are natural anti-nutritional substances, they easily penetrate the cell membranes, disrupt the absorption of iodine by the thyroid gland, and disturb the synthesis of thyroid hormones, they are natural goiter-forming substances. Frequent consumption of cruciferous vegetables, which contain thioglycosides transformed, inter alia, into thiocyanate, is an important element in the prevention of cancer but requires the simultaneous supply of products with a high iodine content.
We have further divided section two into subsections of positive and negative effects of SCN in disease in the context of MPO. The headings and subheadings for Chapter 2 are now as follows:
- - SCN- in Diseases (page 5, line 155)
2.1 – Positive Effect of SCN- in Disease Outcome (page 5, line 170)
2.1.1 – Cardiovascular Disease (page 5, line 171)
2.1.2 – Respiratory Disease (page 6, line 207)
2.1.2.1 – Respiratory Viral Infections (page 7, line 262)
2.2 – Negative Effect of SCN- in Disease Outcome (page 8, line 308)
2.2.1 – Autoimmune Rheumatic Diseases (page 8, line 309)
2.2.2 – Gastrointestinal Disease (page 9, line 355)
We have considered the iodine sequestering function of thiocyanate and while this is important as a limiting factor in utilising thiocyanate as a long-term therapeutic agent - particularly at higher doses, it does detract from the purpose of the review which is to highlight the role of SCN in the pathophysiology of diseases where MPO is implicated. Thus, we feel reviewing this aspect is difficult to relate to the theme of the manuscript.
Quotation - Levels of SCN- in the extracellular fluid of mammals can vary considerably depending on numerous factors, and can reach up to millimolar concentrations for environmentally exposed mucous membranes lining the oral cavity, digestive tract and airway [16, 49]. (last paragraph chapter 2) - what extracellular fluid do we mean? what is this environmental exposure all about?
The above sentence has been revised where we have changed 'extracellular' with 'secreted' and have removed 'environmentally exposed’. We have moved this sentence to Section 1.1.2 where it now has a logical flow (page 3, lines 62-64).
GSH is missing from the abbreviations
We have included GSH as a first-time use abbreviation (page 5, line 152), in the abbreviations list (page 12, line 413), and amended instances of ‘glutathione’ in-text to ‘GSH’ (page 7, lines 227, 236).
I suggest considering carefully the first three sentences of the Abstract; my suggestion (below) and abstract in the article.
Thiocyanate (SCN-) is a pseudohalide anion omnipresent across mammals and is particularly concentrated in secretions within the oral cavity, digestive tract, and airway. Thiocyanate can outcompete chlorine anion (was - free chlorine - whether free chlorine = chloride ions?), and with other halides (Br-, J-, F-) as substrates for myeloperoxidase by undergoing two-electron oxidation with hydrogen peroxide. This forms its respective hypohalous acids (HOX where X- = halides) and in the case of thiocyanate HOSCN, which is also a bactericidal oxidative species involved in the regulation of commensal and pathogenic microflora.
The last sentence of the Abstract - Here, we discuss the therapeutic potential of SCN- for altering disease outcomes across various pathologic processes. My question - why have diseases in which SCN- has no therapeutic effect been also described?
We agree with the reviewer’s comment here and have changed the abstract in accordance to their recommendation (page 1, lines 10, 12-15, 23-24).
Round 2
Reviewer 2 Report
The manuscript is much improved, and the authors have done a thorough job in addressing all of the comments put forward. Just a minor point that the units for rate constants are M-1.s-1, this needs to be corrected (please remove the "k2/"). It would be helpful to readers of this review if there was a summary of the sections subheadings, listed after the abstract and just prior to the body of the article.
Author Response
We have now removed ‘k2/’ for the rate constants (page 5, lines 168-170), as outlined by Reviewer 2.
We also note that ‘SARS-CoV-2’ was not included in the abbreviations list and have now included this (page 12, line 443; highlighted red).
We have now included a Table of Contents containing all Section titles and subheadings (page 1, lines 27-42), as suggested by Reviewer 2.
